# Integrative Metabolic and Transcriptomic Profiling in *Camellia oleifera* and *Camellia meiocarpa* Uncover Potential Mechanisms That Govern Triacylglycerol Degradation during Seed Desiccation

**DOI:** 10.3390/plants12142591

**Published:** 2023-07-08

**Authors:** Mingjie Chen, Yi Zhang, Zhenghua Du, Xiangrui Kong, Xiaofang Zhu

**Affiliations:** 1International Joint Laboratory of Biology and High Value Utilization of Camellia oleifera in Henan Province, College of Life Sciences, Xinyang Normal University, Xinyang 464000, China; 2Center for Horticultural Biology and Metabolomics, Haixia Institute of Science and Technology, Fujian Agriculture and Forestry University, Fuzhou 350002, China; yzhang@email.ncu.edu.cn (Y.Z.); zhenghuadu@fafu.edu.cn (Z.D.); zhuxf819@163.com (X.Z.); 3School of Life Sciences, Nanchang University, Nanchang 330031, China; 4Tea Research Institute, Fujian Academy of Agricultural Sciences, Fuzhou 350012, China; kongxiangrui_2008@163.com; 5Xianyang Jingwei Fu Tea Co., Ltd., Xianyang 712044, China

**Keywords:** *Camellia oleifera*, *Camellia meiocarpa*, fatty acids, oil content, transcriptome, differentially expressed gene, lipid degradation, seed desiccation

## Abstract

Camellia seed oil is a top-end quality of cooking oil in China. The oil quality and quantity are formed during seed maturation and desiccation. So far, it remains largely unresolved whether lipid degradation occurs and contributes to Camellia oil traits. In this study, three different Camellia germplasms, *C. oleifera* cv. *Min 43* (M43), *C. meiocarpa* var. *Qingguo* (QG), and *C. meiocarpa cv Hongguo* (HG) were selected, their seed oil contents and compositions were quantified across different stages of seed desiccation. We found that at the late stage of desiccation, M43 and QG lost a significant portion of seed oil, while such an event was not observed in HG. To explore the molecular bases for the oil loss In M43, the transcriptomic profiling of M43 and HG was performed at the early and the late seed desiccation, respectively, and differentially expressed genes (DEGs) from the lipid metabolic pathway were identified and analyzed. Our data demonstrated that different Camellia species have diverse mechanisms to regulate seed oil accumulation and degradation, and that triacylglycerol-to-terpenoid conversion could account for the oil loss in M43 during late seed desiccation.

## 1. Introduction

*Camellia oleifera* Abel., a member of the Theaceae family, is widely distributed in the subtropical mountain areas of the Yangtze River basin and South China [1,2]. As evergreen broadleaf shrubs or small trees, *Camellia* is tolerant to drought and barren land [3]. *C. oleifera* is mainly used for oil production from its seed kernel, which contains ~50% of oilseed dry mass [4], and oleic acid account for ~80% of its seed oil. Thus, Camellia seed oil is reputed as the ‘‘Oriental olive oil’’ [5]. In addition, the Camellia oil is rich in bioactive compounds, including polyphenol, squalene, and vitamin E, etc. [6]. Thus, Camellia oil shows potent activities of antioxidation, antiaging, anti-inflammation, anticancer, and antibacterial. The long-term consumption of Camellia oil could lower the levels of serum triglycerides and cholesterol, enhance immunity, stabilize blood sugar levels, reduce cardiovascular disease, and prevent hypertension [5].

There are a lot of interests to understand the factors affecting Camellia oil synthesis. Recently, transcriptomic technology has been applied to understand lipid synthesis and response to environmental stresses in *Camellia oleifera* [3,7,8,9,10,11]. Zeng et al. (2014) reported that the mRNA levels of fructose-1,6-bisphosphate aldolase (*CoFBA*) and stearoyl-ACP desaturases (*CoSAD*) are correlated with oil content, whereas fatty acid desaturase 2 (*CoFAD2*) gene expression levels are correlated with fatty acid composition [12]. Feng et al. (2017) analyzed transcriptomic changes during postharvest seed drying [10]. The authors found that natural drying improved the quality and quantity of Camellia seed oil; in addition, putative transcripts were identified that are potentially involved in lipid metabolism during postharvest drying. Lin et al. (2018) performed the transcriptomic profiling of Camellia seeds at four developmental stages, and found that the distribution of the DEG numbers was associated with seed oil accumulation patterns, the authors suggest that the regulation of *CoSAD* and *CoFAD2* is critical to boost oleic acid level at the late stages of seed development [13]. Wu et al. (2019) reported that the coordinated higher expression levels of the upstream gene (*HAD*, *EAR* and *KASI*) are directly associated with increased levels of palmitic acid (C16:0), and continuous higher expression levels of the *SAD* gene could accelerate oleic acid synthesis and accumulation, while coordinated lower expression levels of the downstream genes (*FAD2*, *FAD3*, *FAD7*, *FAD8* and *FAE1*) are correlated with a decreased oleic acid conversion [11]. Gong et al. (2020) applied iso-seq to obtain a full-length transcriptome from *Camellia oleifera* seeds [14]. The authors found that transcript variants could be involved in seed oil biosynthesis. Lin et al. (2022) performed a genome-wide association study, and found that during Camellia domestication, sugar-dependent triacylglycerol lipase 1 (CoSDP1), β-ketoacyl-acyl carrier protein synthase III (KAS III), and CoSAD play important roles in enhancing the yield and quality of Camellia seed oil [15].

For oil seeds such as *B. napus*, *Arabidopsis*, *Crambe abyssinica*, and *Nicotiana tabacum*, lipid contents were reported to decrease during seed maturation [16,17,18]. In *B. napus*, the enzyme activities and abundance of β-oxidation, glyoxylate cycle, and phosphoenolpyruvate carboxykinase were increased during seed desiccation compared with the oil accumulation stage, seed lipid degradation was not associated with net gluconeogenesis activity, and partial lipids were released as CO_2_ [19]. These data suggest that *B. napus* seed oils were actively degraded or converted into other metabolites other than sugars. Zhou et al. (2013) and Liu (2020) reported that harvest time affects seed oil content in *C. oleifera*, and that it remains unclear whether seed oil degradation accounts for it [20,21].

In China, *C. oleifera* is the dominant cultivated oil-tree species [22]. Until now, limited attention has been paid to other Camellia oil-tree species, including *C. meiocarpa* and *C. chekiangoleosa* [10,23]. In this study, two varieties of *C. meiocarpa*, were included, they are: *C. meiocarpa* var. *Hongguo* (HG) and *C. meiocarpa* var. *Qingguo* (QG). The obvious phenotypic difference between these two germplasms is the timing of fruit color changes: on September, HG turns its green fruit color to red when seeds start to mature, while QG keeps the green color of its fruits at same stage (Appendix A). To investigate whether there are species- or variety-specific effects on Camellia seed oil accumulation or degradation, the seeds from HG, QG, and *C. oleifera* cv. *Min 43* (M43) were harvested weekly from late September to mid-November, which span seed maturation and desiccation. Seed oil contents and compositions were then measured and compared. We found that at the late stage of desiccation, M43 and QG lost a significant portion of seed oil, while such oil loss was not observed in HG. To elucidate the underlying molecular bases, transcriptomic profiling from M43 and HG was performed at the early (26 September, 47 weeks after anthesis) and the late (1 November, 52 weeks after anthesis) desiccation stages, respectively, and differentially expressed genes (DEGs) in the lipid metabolic pathway were identified and analyzed. Our data provide novel insights about the potential mechanisms responsible for seed oil degradation in M43 during preharvest seed desiccation stages.

## 2. Results

### 2.1. Camellia oleifera and Camellia meiocarpa Showed Diverse Oil Accumulation Trends during Preharvest Seed Desiccation Stages

Three *Camellia* germplasms, including *C. oleifera* cv. *Min 43* (M43), *C. meiocarpa* var. *Hongguo* (HG), and *C. meiocarpa* var. *Qingguo* (QG), were planted side by side in a state forest farm the in Fujian province of China. Camellia seed oil accumulation during the entire period of seed development had been surveyed before [13,14]. Like other oil crops, Camellia seed maturation varies with germplasms and growth locations. In this study, we are interested to understand Camellia seed oil accumulation during seed desiccation, and so the fruits from three different Camellia germplasms were harvested weekly, spaning the 46th to 54th week after anthesis. When fruits were harvested on the 47th week after anthesis, the seed coats from QG showed a darker color compared with M43 and HG (Appendix A), suggesting that the QG fruits reached maturity slightly earlier than that of M43 and HG. HG showed similar developmental timelines as M43. GC-MS analysis demonstrated that the seed oil from these three germplasms possess a similar fatty acid profile, and ten fatty acids were identified, including: palmitic acid (16:0), cis-9-hexadecenoic acid (16:1^Δ9^), cis-11-hexadecenoic acid (16:1^Δ11^), stearic acid (18:0), oleic acid (18:1^Δ9^), cis-11-octadecenoic acid (18:1^Δ11^), linoleic acid (18:2^Δ9,12^), linolenic acid (18:3^Δ9,12,15^), arachidic acid (20:0), and paullinic acid (20:1^Δ13^) (Appendix A).

The seed oil contents among these three germplasms showed diverse accumulation patterns during seed maturation and desiccation: at the 46th week after anthesis, both QG and HG had similar seed oil contents; in the following week, both showed large differences in oil accumulation rates where the oil content of HG increased rapidly, followed by a much gentle increase. In contrast, the seed oil content from QG showed a slower but steady increase until the 53rd week, followed by a sharp decrease. From the 53rd to the 54th week, QG lost ~25% of its seed oil reserve. The seed oil content from M43 showed biphasic on the 49th and the 52nd week, respectively. From the 52nd to 54th week, M43 lost ~30% of its seed oil reserve. In contrast, at the same period, the seed oil content of HG even slightly increased. Despite their diverse oil accumulation patterns, these three cultivars showed similar seed oil contents on the 54th week after anthesis (Figure 1).

### 2.2. The Accumulation Patterns of Individual Fatty Acid Species during Seed Desiccation

The contents of individual fatty acid species were quantified during various stages of seed desiccation; the data were presented in Figure 2 and Appendix A. The oleic acid, the dominant fatty acid, showed diverse changing patterns among these three germplasms (Figure 2). In QG, the oleic acid showed an almost linear increase from the 46th to the 53rd week after anthesis, then followed by a 26.4% decrease on the 54th week after anthesis (Figure 2, left upper panel). In HG, the oleic acid showed a rapid increase from the 46th to the 47th week after anthesis, then remained almost stagnant until the 49th week. At the 50th week, the oleic acid (18:1^Δ9^) slightly decreased, followed by a constant increase until the 54th week. Interestingly, on the 50th week, the drop in 18:1^Δ9^ is accompanied by the concurrent increase in linoleic acid, suggesting that the decline in oleic acid could be due to its accelerated conversion into linoleic acid at this time point (Figure 2, middle upper panel). In M43, the oleic acid showed two peaks, with the first small peak presented on the 49th week, the second large peak presented on the 52nd week, then followed by a 26.7% decrease on the 54th week after anthesis. The linoleic acid content did not show coordinated changes with oleic acid as we saw in HG (Figure 2, right upper panel). The linoleic acid (18:2) is the major polyunsaturated fatty acid, and accounts for 3.9% to 4.4% of seed dry mass; palmitic acid (16:0) is the dominant saturated fatty acid in Camellia seed oil, and accounts for 3.3% to 4.0% of seed dry mass. During seed desiccation, the linoleic acid and palmitic acid contents kept fairly constant (Figure 2, upper panel).

The oil contents for steric acid (18:0), cis-11-octadecenoic acid (18:1^Δ11^), linolenic acid (18:3), and arachidic acid (20:0) were in the range of 0.1–1.4% of seed dry mass. Beginning on the 50th week, 18:0 showed a general upward trend, while 18:1^Δ11^ showed a downward trend from these three germplasms. Beginning on the 52nd week, the 18:0 contents from QG and M43 started to decrease, while its content in HG kept rising. The contents for 18:3 and 20:0 only showed small variations during seed desiccation (Figure 2 middle panel). Cis-11-hexadecenoic acid (16:1^Δ11^) and paullinic acid (20:1) accounted for less than 0.1% of seed dry mass (Figure 2, lower panel). Their variations would not significantly affect the total seed oil contents.

Next, we expressed the seed oil content data as mol percent (Mol%), then examined fatty acid compositional changes during seed desiccation. The proportion of oleic acid from QG and M43 showed general increasing trends with seed desiccation. Meanwhile, the relative proportion of palmitic acid and linoleic acid gradually decreased (Figure 3). The oleic acid proportion from HG showed two peaks, with the first peak appearing on the 48th week after anthesis, followed by a decrease until the 50th week, then increase again until the 54th week after anthesis. Accordingly, the linoleic acid proportion showed an opposite changing trend to that of oleic acid (Figure 3, middle panel). The other six fatty acids were minor components, and only showed small variations during seed desiccation (Figure 3).

The unsaturated fatty acid (USFA)-to-saturated fatty acid (SFA) ratios showed a generally increasing trend during desiccation among the three germplasms (Figure 4, upper panel). The monounsaturated fatty acid (MUFA)-to-polyunsaturated fatty acid (PUFA) ratios from HG showed large variations during seed desiccation, with the first peak appearing on the 48th week after anthesis, then reached to nadir on the 50th week. After that, the ratio kept increasing until the 54th week. In contrast, QG and M43 showed a stable increase until the 53rd week (Figure 4, middle panel). The ratios of 18C to 16C fatty acid shared similar changing trends as the USFA-to-SFA ratios (Figure 4, lower panel), largely due to the fact that oleic acid (18:1^Δ9^) and palmitic acid (16:0) are the dominant USFA and SFA, respectively.

### 2.3. The Correlations among Total Seed Oil Contents and Individual Fatty Acid Species

Fatty acids are the major components in Camellia seed oil (triacylglycerol, TAG). To better understand the relationships between oil contents and fatty acid species, below, we briefly describe the major steps and enzymes for fatty acid synthesis (Figure 5a). In plastid, beta-ketoacyl-ACP synthase I (KAS I) uses acetyl-CoA (Ac-CoA) as substrate to synthesize fatty acids with a chain length less than 16C, then KAS II converts 16C fatty acid (16:0) into 18C fatty acid (18:0) [24,25]. 18:0 is either exported into cytoplasm or desaturated to 18:1 by plastidial steryl-ACP desaturase (SAD) before exportation [26]. After exportation, partial 18C fatty acid is converted to 20C fatty acids by the endoplasmic reticulum (ER)-associated fatty acid elongase complex (FAE) [27]. Thus, almost all the 18C and 20C fatty acids are synthesized through KAS II except 18:1^Δ11^ and 20:1^Δ13^, which are synthesized by FAE in ER (Figure 5a). FatB converts 16:0-ACP into 16:0-CoA in plastid and determines 16:0 fate for TAG synthesis in ER [28,29]. FatA is involved in the export of 18:0-ACP or 18:1-ACP to cytoplasm for oil synthesis. FAD2 desaturates 18:1 into 18:2 in ER [30,31], then FAD3 further desaturates 18:2 into 18:3 [32]. In cytoplasm, FAE elongates cis-9-hexadecenoic acid (16:1^Δ9^) into cis-11-octadecenoic acid (18:1^Δ11^) [27,33], which is further elongated into paullinic acid (20:1^Δ13^).

The correlations among total seed oil contents and individual fatty acid contents were analyzed (Appendix A). In all three germplasms, total seed oil contents were significantly and positively correlated with the contents of 18:0, 18:1^Δ9^, and 18:3, but not correlated with the contents of 16:1^Δ11^, 20:0, and 18:2 (Figure 5b–d). In both of the *C. meiocarpa* germplasms (QG and HG), 20:1 contents were also positively correlated with total seed oil contents (Figure 5b,c), while no such correlation was observed in *C. oleifera* cv. *Min 43* (Figure 5d). In QG and M43, total seed oil contents were also positively correlated with 16:0 contents (Figure 5b,d), while no such correlation was observed in HG (Figure 5c).

Among individual fatty acid species, both positive and negative correlations were identified. In all three germplasms, 18:1^Δ9^ is positively correlated with 18:0, 18:3, and 20:1 (Figure 5b–d); the positive correlation between 18:2 and 18:3 was only observed in QG (Figure 5b). In both varieties of *C. meiocarpa*, 18:1^Δ11^ was negatively correlated with 18:0 (Figure 5b,c), while such correlation was not found in M43 (Figure 5d).

### 2.4. The Synchronization of Enzyme Activity Fluctuation during Seed Desiccation

As described in Figure 5a, all the 18C and 20C fatty acids (except 18:1^Δ11^ and 20:1^Δ13^) are synthesized through KAS II. When the absolute contents of 18C and 20C fatty acids (except 18:1^Δ11^ and 20:1^Δ13^) are added up at each sampling week, the difference relative to the previous week reflect the average activity changes of KAS II during that week in planta. Similarly, the palmitic acid content changes relative to the previous week represent FatB activity changes. Since all the 18:1^Δ9^, 18:2, and 18:3 are synthesized through SAD, and 18:2 and 18:3 are synthesized through FAD2, the total seed content changes of 18:1^Δ9^, 18:2, and 18:3 relative to the previous week reflect SAD activity changes. The total seed content changes of 18:2 and 18:3 relative to the previous week reflect FAD2 activity changes, while the 18:3 content changes relative to the previous week reflect FAD3 activity changes. The total content changes of cis-11-octadecenoic acid (18:1^Δ11^) and paullinic acid (20:1^Δ13^) relative to the previous week reflect FAE activity changes in planta (Figure 5a). Based on the absolute seed fatty acid contents as presented in Figure 2 and Appendix A, the weekly average enzyme activity fluctuations for KAS II, SAD, FatB, FAD2, FAD3, and FAE were plotted and presented in Figure 6. In QG and M43, all these enzymes showed highly synchronized fluctuation patterns (Figure 6, left and middle panel). In HG, these synchronized fluctuation patterns were sustained until the 50th week after anthesis, after that, FAD2 showed discordant change trends with KAS II and SAD (Figure 6, right panel). Considering that QG and M43 showed rapid oil accumulation, whereas the oil accumulation in HG was much slower (Figure 1), we reasoned that these concerted enzyme activity fluctuations in fatty acid synthesis could maximize seed oil biosynthetic efficiency.

### 2.5. RNA-seq, De Novo Transcriptome Assembly, and Functional Annotation

As described above, on the 47th week after anthesis, M43 and HG contained similar levels of seed oil; on the 52nd week after anthesis, the seed oil content of M43 was significantly higher than that of HG, then decreased dramatically, while such a reduction was not observed in HG (Figure 1). To explore the potential mechanisms governing seed oil loss in M43 at the late phase of seed desiccation, seeds were harvested on the 47th and the 52nd week after anthesis for transcriptomic analysis.

The four libraries of HG contained 38.99 G clean reads, and bioinformatics analysis identified 46,620 unigenes (Appendix A), which contain 32,552 protein coding sequences (CDS) (Appendix A). The four libraries of M43 contained 41.49 G clean reads, and bioinformatics analysis identified 52,997 unigenes (Appendix A), which contain 34,930 protein coding sequences (Appendix A). To identify homologous genes between M43 and HG, their protein sequences were blasted to each other by using reciprocal best BLAST hit (RBH) method, and ~30,000 homologous genes were identified between M43 and HG (Appendix A).

### 2.6. Identification of Candidate DEGs Involved in Seed Oil Degradation in C. oleifera cv. Min 43

Compared to the 47th week after anthesis, on the 52nd week, 5940 and 7644 unigenes were differentially expressed from HG and M43, respectively (Appendix A). GO term enrichment analysis demonstrated that in HG, the downregulated unigenes are enriched in monooxygenase activity, oxidoreductase activity, and toxin activity (Appendix A), whereas the upregulated unigenes are enriched in response to chitin (Appendix A). In M43, the downregulated unigenes are enriched in photosystem I stabilization (Appendix A), while the upregulated unigenes are enriched in chitin binding, chitinase activity and monooxygenase activity, defense response, response to wounding, response to chitin, camalexin biosynthetic process, and cellular response to sulfur starvation (Appendix A).

The genes involved in lipid pathway were further screened. In total, 382 homologous unigenes for lipid metabolism were functionally annotated (Appendix A). Within lipid pathway, the DEGs between the 47th and the 52nd week samples were screened. In total, 75 and 107 DEGs were identified from HG and M43, respectively (Appendix A). Comparing to the 47th week after anthesis, in HG, 29 out of the 75 genes were upregulated and 46 genes downregulated; in M43, 60 of the 107 genes were upregulated and 47 genes downregulated (Appendix A).

To identify the potential genes involved in the TAG degradation of M43, we reasoned that those putative genes would exhibit relatively higher expression levels in M43 compared with that of HG on the 52nd week after anthesis. Thus, we calculated the gene expression fold changes of M43 relative to HG by dividing their respective expression ratios. By using >2 folds’ changes as cutoff, 58 candidate genes were identified (Table 1). Based on respective expression patterns within individual germplasms, these genes can be divided into five groups. Group I includes six genes, which were upregulated from M43 and HG during seed desiccation, and M43 showed a much higher upregulation compared with that of HG. Group II includes five genes, which were downregulated in M43 and HG during seed desiccation, and M43 showed a smaller reduction compared with HG. Group III includes six genes, which were upregulated in M43 but downregulated in HG. One gene (TRINITY_DN21501_c0_g1_i1_1) is involved in steroid biosynthesis. Group IV includes 15 genes, which kept constant in M43 but downregulated in HG. Within this group, four genes are involved in fatty acid or glycan degradation, they are: TRINITY_DN23188_c1_g1_i1_1, TRINITY_DN28894_c0_g2_i1_1, TRINITY_DN29729_c0_g2_i1_1 and TRINITY_DN26102_c2_g4_i1_1; and two genes are involved in steroid biosynthesis, including TRINITY_DN23273_c1_g1_i1_1 and TRINITY_DN27207_c0_g1_i3_1 (Table 1). Finally, group V includes 26 genes, which were upregulated in M43, but kept constant in HG. Five of them are involved in steroid biosynthesis, including: TRINITY_DN17413_c0_g1_i1_2, TRINITY_DN18365_c0_g2_i1_1, TRINITY_DN26549_c2_g1_i1_1, TRINITY_DN28994_c1_g1_i1_1, and TRINITY_DN4285_c0_g1_i1_2 (Table 1).

### 2.7. qRT-PCR Validated RNA-seq Results

To confirm the RNA-seq results, at least one DEG was selected from each group in Table 1. A total of eight DEGs (Appendix A) were selected and subjected to qRT-PCR analysis, then the results were compared with the RNA-seq data. We found that both datasets showed generally similar changing trends (Figure 7).

## 3. Discussion

### 3.1. C. meiocarpa and C. oleifera Accumulate High Levels of Oleic Acid

Both *C. meiocarpa* and *C. oleifera* accumulated 65% to 72% oleic acid (Figure 3), it is interesting to understand why Camellia seed kernel preferably accumulates oleic acid to such high levels. In hickory seeds, a high level of *SAD* with a low level of *FAD2* expression was associated with high levels of oleic acid accumulation [34]. In olive, the expression of *FAD2* is repressed by a newly evolved small RNA, and contributes to its high-level oleic acid content [35]. Zeng et al. (2014) showed that in Camellia seeds, fructose-1,6-bisphosphate aldolase (*CoFBA*) and *CoSAD* mRNA levels were well-correlated with oil content, whereas *CoFAD2* gene expression levels were correlated with fatty acid composition [12]. The authors propose that *CoFBA* and *CoSAD* are two important factors to determine Camellia seed oil content, while *CoFAD2* gene expression levels were correlated with fatty acid composition. In this study, we found that during seed maturation and desiccation, the coordinated changes in enzyme activities of KAS II, SAD, FatB, FAD2, FAD3, and FAE were associated with seed oil accumulation rates (Figure 1 and Figure 6). Correlation analysis demonstrated that seed oil contents were significantly and positively correlated with 18:0, 18:1^Δ9^, and 18:3 (Figure 5b–d). Notably, the 18C fatty acids were eight-folds higher than that of 16C fatty acids (Figure 4, lower panel), suggesting that KAS II, a key enzyme to convert 16C to 18C fatty acids, would play critical roles for high oleic acid accumulation in both *C. oleifera* and *C. meiocarpa*. Besides KAS II, FatA, SAD, and FAD2 are located on the critical positions of the fatty acid synthesis pathway; thus, they are expected to play important roles for high oleic acid accumulation. Consistent with this notion, Gong et al. (2020) reported that *SAD* and *FAD2* showed high levels of expression during seed maturation [14]. In Arabidopsis, several plastidial stearoyl-ACP desaturase (SAD) paralogs, including FAB2, AAD1, AAD5, and AAD6, work redundantly to catalyze 18:0-ACP into 18:1-ACP during seed maturation [36,37,38]. This could also be the case in Camellia seeds, considering that 18:3 only accounts for a minor fraction of seed oil (Figure 3), thereby suggesting that FAD3 enzymes are either not active or their genes are expressed at very low levels, thus restricting 18:1^Δ9^ from being converted into 18:2 and contributing to high oleic acid accumulation. FAD6, FAD7, and FAD8 are plastidial galactolipid desaturases, which sequentially desaturate 18:1 (16:1) to 18:2 (16:2) to 18:3 (16:3) [39,40,41]. *Camellia* are 18:3 plants, their leaf galactolipids are dominated by 18:3 fatty acids [42,43], suggesting that FAD6, FAD7, and FAD8 play important roles for galactolipid desaturation in Camellia leaf tissue. However, Camellia seed oil only contains trace amounts of 18:3, while 16:2 or 16:3 fatty acids are barely detectable (Figure 2 and Appendix A). These data suggest that FAD6, FAD7, and FAD8 do not make significant contributions to Camellia seed oil synthesis.

### 3.2. The Exocarp Photosynthesis May Contribute to Camellia Seed Oil Accumulation

During seed maturation and desiccation, HG exhibited distinct seed oil accumulation patterns that differentiate itself from QG and M43 (Figure 1). In view of its roles in the trophic connection between seeds and the mother plant, green exocarp could possess photosynthetic capacity and contribute to seed oil accumulation. In chloroembryos such as oilseed rape, seed photosynthesis plays a role in the accumulation of storage lipids [44,45]. The Camellia embryo is embedded in fruit peels, including exocarp, mesocarp, and endocarp. With seed maturation, the endocarp lignifies into a dark seed coat. The seed coat in combination with mesocarp would prevent light penetration into the embryo (Appendix A). Thus, unlike chloroembryos, the photosynthesis from Camellia embryo would not make any significant contributions to seed oil accumulation. However, the fruit exocarp contains chlorophyll (Appendix A) and stomata [46], and has the potential to assimilate atmospheric CO_2_ by photosynthesis to fuel Camellia seed oil accumulation. In accordance with this notion, it was reported that tomato fruit photosynthesis is restricted to the green phases of development, and contributes to net sugar accumulation and growth of the fruits [47,48]. QG and M43 showed green exocarp during seed maturation and desiccation; in contrast, the exocarp of HG turned red in color at the early phase of seed desiccation, which may reduce its photosynthetic capacity of the exocarp. In accordance with the fruit color changes, the seed oil accumulation from HG was much slower compared with QG or M43 (Figure 1). These observations suggest that the exocarp photosynthesis could be an important factor that affects seed oil accumulation in Camellia seeds.

### 3.3. The Harvest Time of Camellia Seeds Is Important to Obtain Higher Oil Yield

In this study, we found a pronounced seed oil loss from QG and M43 at the late phase of seed desiccation (Figure 1), and most individual fatty acid contents also reduced concurrently (Figure 2). In contrast, the total oil contents from HG was not reduced, instead, 18:1^Δ9^ content continually increased at the expense of 16:0 and 18:2 reductions (Figure 2, upper middle panel). Although QG and M43 accumulated much higher seed oils than that of HG between the 51st and the 53rd week after anthesis, these three germplasms contained similar levels of seed oil (~43% of seed dry mass) when their seeds were harvested on the 54th week (Figure 1). Thus, the seed oil accumulation from individual germplasms in the Theaceae family is diverse and highly dynamic, and harvesting the seeds at the right time is therefore important to obtain high oil yield.

### 3.4. The Potential Pathways Governing Triacylglycerol Degradation at the Late Phase of Seed Desiccation

Previous studies demonstrated that triacylglycerol synthesis and degradation are two concurrent processes during oil seed filling [19,49]. During seed desiccation, the Camellia seed endocarp is lignified into a seed coat, which could prevent nutrients from freely diffusing from the mother plant to the seed kernel. However, some biosynthetic processes in the seed could still persist as other oil seeds. For example, in *B. napus*, the enzymes of β-oxidation, the glyoxylate cycle, and phosphoenolpyruvate carboxykinase were present in embryos during oil accumulation, and increased in activities and abundance with seed maturation and desiccation [10,17,50]. In this study, we performed transcriptomic profiling at the early and the late phases of seed desiccation from HG and M43, and identified four genes in fatty acid or glycan degradation pathway (Table 1). These data suggest that the transcriptional upregulation of fatty acid degradation genes could play an important role in governing seed oil degradation in M43. In Arabidopsis and rape, triacylglycerol lipases (SDP1 and SFAR) have been demonstrated to be involved in TAG degradation during seed maturation [51,52]. Surprisingly, these lipase homologs were not present from the list in Table 1. We speculate that there are two potential explanations: (1) our selection criteria for DEGs could be too harsh such that some candidate genes could be missed out and/or (2) Arabidopsis or rape seeds are type I seeds, which are small and tolerant to desiccation. In contrast, Camellia seeds are type II seeds, which are large and only partially tolerant to desiccation [53,54]. It remains an open question whether these two types of oil seeds could leverage different lipase for TAG degradation during seed development.

Chia et al. (2005) demonstrated that seed oil degradation in *B. napus* could be fueled into seed protein synthesis [19]. In M43, eight genes in terpenoid (steroid/sesquiterpenoid/triterpenoid) synthesis pathway were upregulated (Table 1). This raises the possibility that seed oil degradation in M43 could be channeled into terpenoid synthesis. It has been reported that during Camellia seed maturation, seed saponin continuously increased to more than 20% of its seed dry mass [20,55,56]. Our data provide potential molecular evidences that TAG degradation could be associated with terpenoid accumulation at the late phase of seed desiccation.

### 3.5. The Physiological Implications of Camellia TAG Degradation at the Late Phase of Seed Desiccation

As we discussed above, Camellia seeds are only partially tolerant to desiccation [53,54]; thus, mature Camellia seeds have a relatively short time window to germinate under natural drought conditions. We speculate that the oil degradation at the late phase of seed desiccation could make Camellia seeds better prepared for seed germination and seedling establishment and thus may confer evolutionary fitness for Camellia species. This notion is in accordance with a previous report that ripe tea seeds germinated quickly and showed a high rate of germination, while the unripe seeds germinated slowly and unevenly, and showed relatively low germination rates [57]. Besides facilitating seed germination, the saponins synthesized at the late phase of seed desiccation could have functions in the deterrence of animal predators or inhibition of pathogen infection, thereby enhancing seed survival and spreading.

## 4. Materials and Methods

### 4.1. Plant Materials and Growth Conditions

*Camellia oleifera* cv. *Min 43*, developed by Fujian Academy of Forestry in 1985, is a colonial propagated elite cultivar; *Camellia meiocarpa* var. *Qingguo* and *Camellia meiocarpa* var. *Hongguo* are two elite landraces in the Fujian province of China. These three germplasms were planted side by side in a state forest farm for ~10 years, which is located in Tongkou, Minhou County, Fujian province (26°05′ N, 119°17′ E) (Appendix A). *C. oleifera* and *C. meiocarpa* fruits show similar developmental timelines: both blossom and fertilize in late October or early November, the fertilized fruits stay in dormancy during the winter, then start expansion and dry mass accumulation the following May. In September, the average seed weight reaches a constant, the water content continues to decline and is accompanied with dry mass increase, and the seed enters the desiccation stage. At early to mid-November, the fruits are harvested for oil production. In this study, the fruits were harvested weekly from 20 September to 14 November 2015, which correspond to the 46th to the 54th week after anthesis. At each sampling week, the harvested fruits were selected based on the following two criteria: (1) the fruits were positioned on the same direction of the tree so that they would likely receive similar amounts of sunlight and (2) the fruits were uniform in size and color appearance within individual germplasms. Thirty fruits were harvested at 10:00 AM from each germplasm, the mesocarp and seed coat were removed and the kernels from the same germplasm were mixed and randomly divided into six parts, with each part containing 20 seeds as one replicate. Three parts were frozen in liquid nitrogen, then stored in −80 °C freezer for total RNA extraction; the other three parts were cut into flakes with a blade, then freeze-dried (Labcoon, KS, USA) for oil content measurement.

### 4.2. Seed Oil Content Analysis

Kernel powder (~10 mg) was methylated in 1.0 mL of methanol containing 5% sulfuric acid (*v*/*v*); 75 μg margaric acid was added as internal standard. After heating at 95 °C in a dry bath for 90 min, the fatty acid methyl esters (FAMEs) were extracted into hexane, concentrated under gentle nitrogen stream, then injected into gas chromatography mass spectrometry in a split ratio of 20:1. The FAMEs were separated in RT-2560 column (0.25 mm × 30 m × 0.25 μm, RESTEK, Bellefonte, PA, USA) and detected by flame ionization detector (FID) and quadrupole mass analyzer (MS) (MDGC/GCMS-2010, Shimadzu, Kyoto, Japan), respectively. Helium flow rates for MS and FID were 1.0 mL min^−1^. The temperature for the injection port, ion source, and detector was 240 °C, 200 °C, and 220 °C, respectively. The flow rates of hydrogen, nitrogen, and zero air were 40, 30, and 400 mL min^−1^, respectively. After 5 min of solvent delay at 150 °C, oven temperature was ramped at 2 °C min^−1^ to 200 °C, held for 5 min, then ramped to 150 °C for next sample injection. The fatty acids were identified from MS data by Quest NIST14 chemical library, and quantified from FID data by normalizing to the peak area of internal standard. The kernel oil content was calculated by dividing the total FAMEs to kernel dry mass weight; the data were also transformed into mole percent to represent oil fatty acid composition. At each sampling point, triplicates were used for fatty acid analysis.

### 4.3. Total RNA Extraction

Total RNA was isolated from kernel by a modified CTAB method [58]. Kernel was first ground into powder in the presence of liquid nitrogen, 1.5 g of powder was transferred into 50 mL centrifuge tube, and then 10 mL of CTAB solution and 450 μL of β-mercaptoethanol were added, mixed, heated in 65 °C water bath for 20 min, and then centrifuged at 12,000× *g* for 10 min (4 °C). The supernatant was transferred into a prechilled tube containing equal volumes of chloroform/isopropanol (24:1, *v*/*v*). The mixture was incubated in ice-water bath for 10 min, then centrifuged at 12,000× *g* for 15 min (4 °C). The above extraction steps were repeated, the supernatant was transferred into a clean tube, and half of a volume of prechilled 8 M LiCl solution and 1% β-mercaptoethanol were added, mixed, then stored in −20 °C freezer overnight. The next day, after centrifugation, the pellet was washed twice with prechilled 75% ethanol, then air dried. The dried pellet was dissolved in 300 μL DEPC-treated water and stored in −80 °C freezer. The quality and concentration of the total RNA was analyzed by 1% agarose gel electrophoresis (Appendix A) and spectrophotometer (NanoDrop 2000, Thermo Scientific, Pleasanton, CA, USA), respectively.

### 4.4. Library Construction and Transcriptome Sequencing

Based on their oil accumulation patterns, seeds harvested on the 47th and the 52nd week after anthesis from HG and M43 were selected for transcriptomic profiling, at each sampling point, two replicates were used. For library construction, mRNA was purified from 6.0 μg of total RNA by oligo-dT-bound magnetic beads, the sequencing libraries were constructed by using NEBNext Ultra^TM^ RNA Library Prep Kit (Illumina, NEB, San Diego, CA, USA) and following the manufacturer’s instructions. The libraries were sequenced on an Illumina Hiseq X platform, 150 bp paired-end reads were generated.

### 4.5. Bioinformatics Analyses

Raw reads were processed using Trimmomatic [59]; the reads containing poly-N and the low-quality reads were removed. After removing adaptor and low-quality sequences, the clean reads were assembled into expressed sequence tag clusters (contigs), then de novo assembled into transcripts by using Trinity in paired-end method [60].

The function of the unigenes was annotated by alignment with a database (NCBI non-redundant (NR), KOG, GO, SwissProt, eggNOG, and KEGG) by Blastx, the threshold E-value was set at 10^−5^ [61,62]. The proteins with the highest hits to the unigenes were used to assign functional annotations.

Total reads per kilobases per million reads (FPKM) [63] and read count values of each unigene were calculated using bowtie 2 [64] and eXpress [65]. DEGs were identified using the DESeq [66] functions estimate size factors and nbinom test. *p* value < 0.05 and fold change > 2 or fold change < 0.5 were set as the threshold for significantly differential expression. The comparative transcriptome analysis was performed using a workflow modified from a previous study [67]. Briefly, the protein sequence of M43 and HG were blast to each other to identify homologous genes by using reciprocal best BLAST hit (RBH) method. Gene ontology (GO) classification was performed by the mapping relation between SwissProt and GO term. The unigenes were mapped to the Kyoto Encyclopedia of Genes and Genomes database (KEGG) to annotate their potential metabolic pathways.

### 4.6. Quantitative RT-PCR Analysis

qRT-PCR was applied to validate RNA-seq results. The names, accession numbers, and primer sequences of the selected eight genes were provided in Appendix A. Two micrograms of total RNA were reverse-transcribed by M-uLV reverse transcriptase to obtain cDNA, then qPCR was performed in a 20 μL volume by using a SYBR Premix Es Taq kit (with Tli RNase H) (Mei5 Biotechnology, Beijing, China). The PCR program was: initial denaturation was at 95 °C for 30 s, followed by 40 cycles of 95 °C for 15 s and 55 °C for 15 s; the PCR was finished by 1 cycle of 95 °C for 60 s, 55 °C for 30 s, and 72 °C for 30 s. *CoGAPDH* was used as the reference gene. Three biological replicates and three technical replicates were performed. Fold difference was calculated using 2^−ΔΔCt^ method [68].

### 4.7. Correlation Analysis

Multiple regression analysis was performed by SPSS (V17.0; SPSS, IBM, Armonk, NY, USA). The significance of correlations between different parameters were determined by bivariate correlations based on Pearson’s correlation (two-tailed).

## 5. Conclusions

During seed desiccation, the oil accumulation in *C. oleifera* and *C. meiocarpa* showed diverse trends; the fatty acid compositions were continuously modified with the increased ratios of unsaturated fatty acids to saturated fatty acids and monounsaturated fatty acids to polyunsaturated fatty acids. At the late phase of seed desiccation, *C. oleifera* cv. *Min 43* and *C. meiocarpa* var. *Qingguo* lost more than 25% of their seed oil reserves, while such an event was not observed in *C. meiocarpa* var. *Hongguo*. Transcriptomic profiling between *C. oleifera* cv. *Min 43* and *C. meiocarpa* var. *Hongguo* suggests that partial oils in M43 were degraded and could be fed into terpenoid synthesis at the late phase of seed desiccation.

## Figures and Tables

**Figure 1 plants-12-02591-f001:**
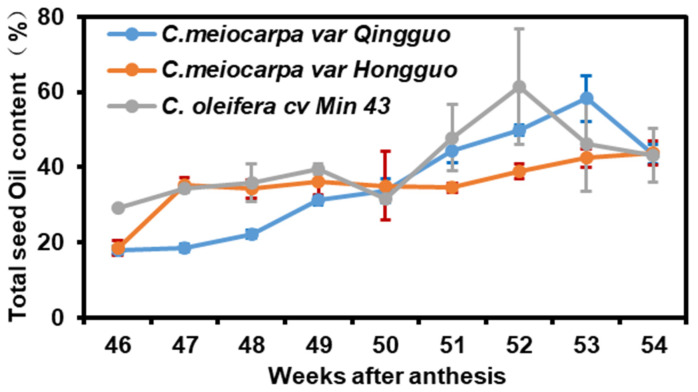
The total seed oil contents during seed desiccation. Data were expressed as mean ± SD (*n* = 3).

**Figure 2 plants-12-02591-f002:**
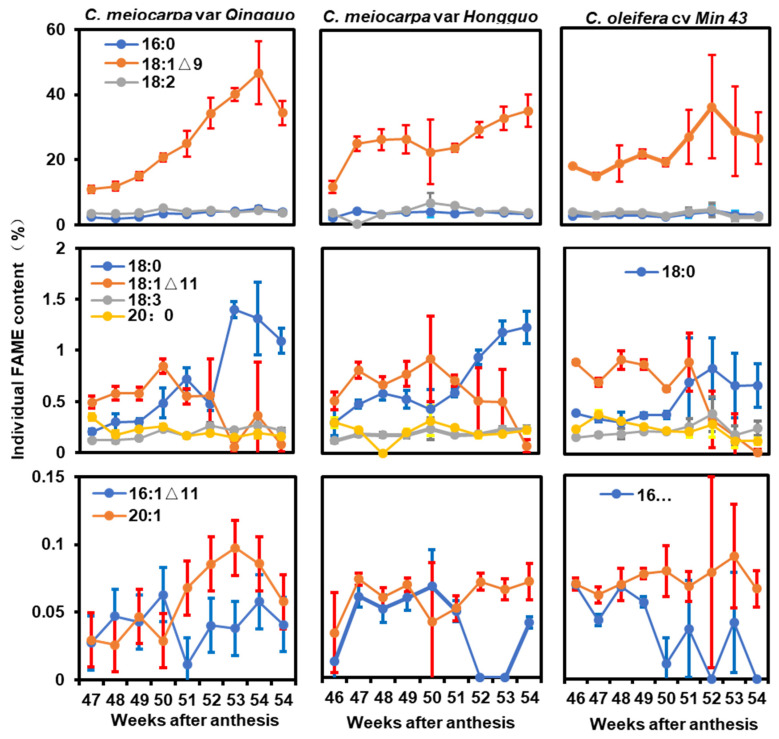
Individual fatty acid content changes during seed desiccation in *C. meiocarpa* var. *Qingguo*, *C. meiocarpa* var. *Hongguo* and *C. oleifera* cv. *Min 43* during seed desiccation. Data were expressed as mean ± SD (*n* = 3).

**Figure 3 plants-12-02591-f003:**
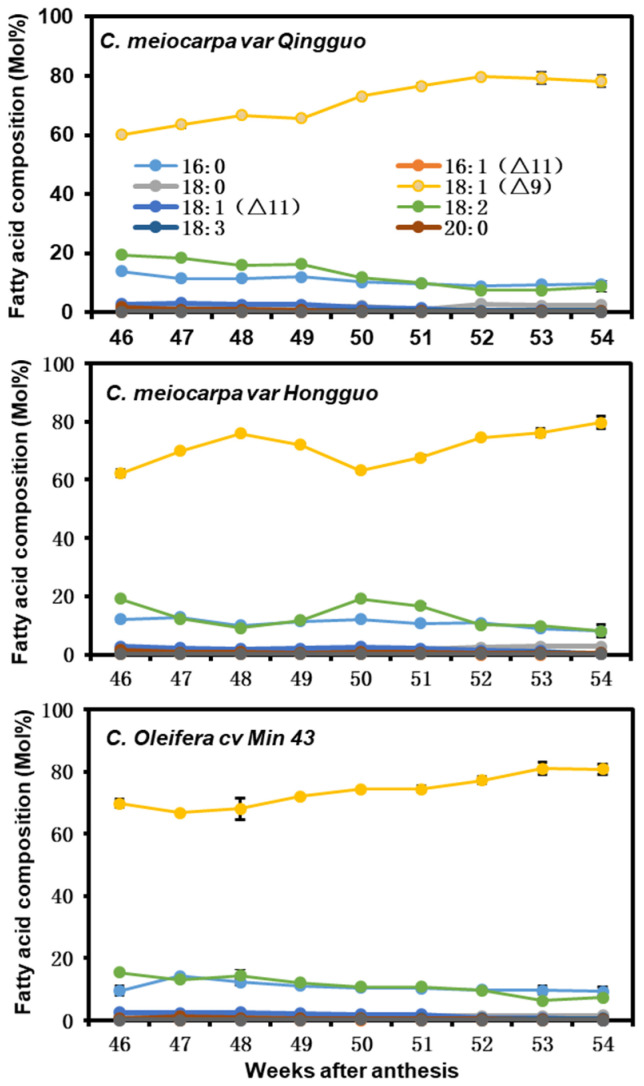
The seed fatty acid compositional changes during seed desiccation. Data were expressed as mean ± SD (*n* = 3).

**Figure 4 plants-12-02591-f004:**
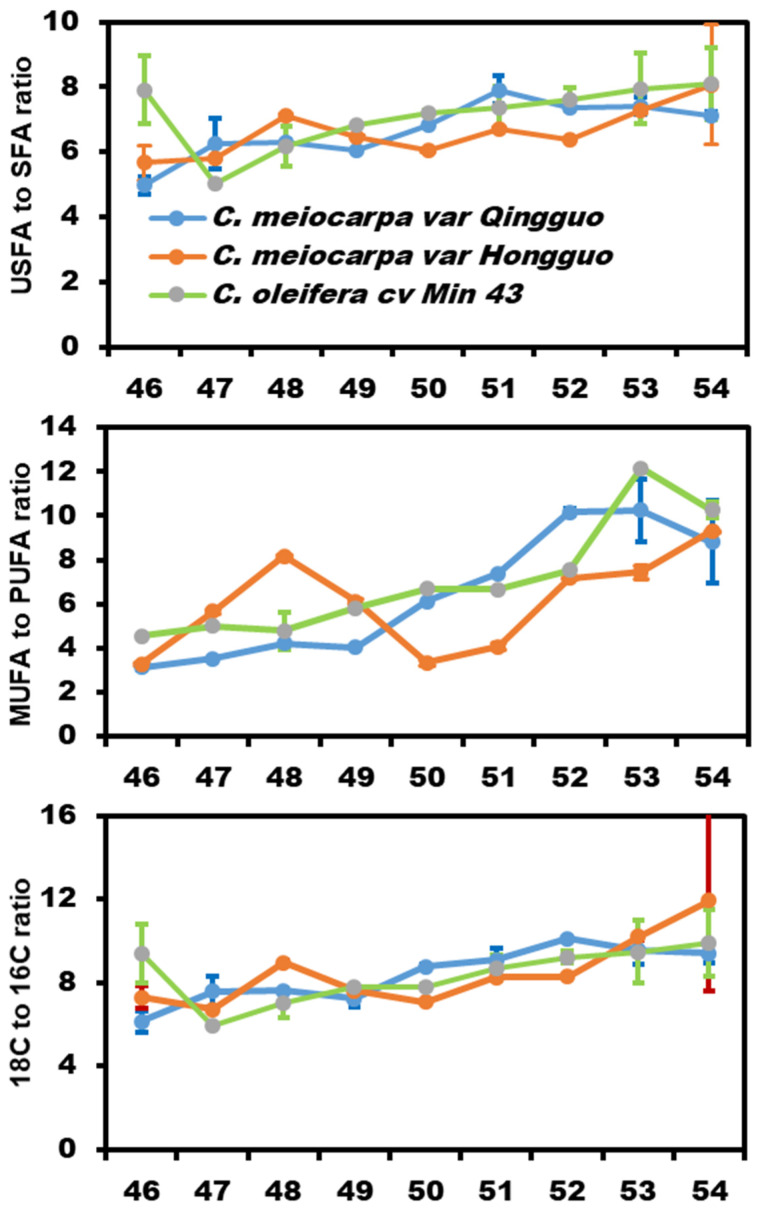
The seed oil compositional changes in *C. meiocarpa* var. *Qingguo*, *C. meiocarpa* var. *Hongguo,* and *C. oleifera* cv. *Min 43* during seed maturation and desiccation. Data were expressed as mean ± SD (*n* = 3). USFA: Unsaturated fatty acid; SFA: saturated fatty acid; MUFA: monounsaturated fatty acid; PUFA: polyunsaturated fatty acid.

**Figure 5 plants-12-02591-f005:**
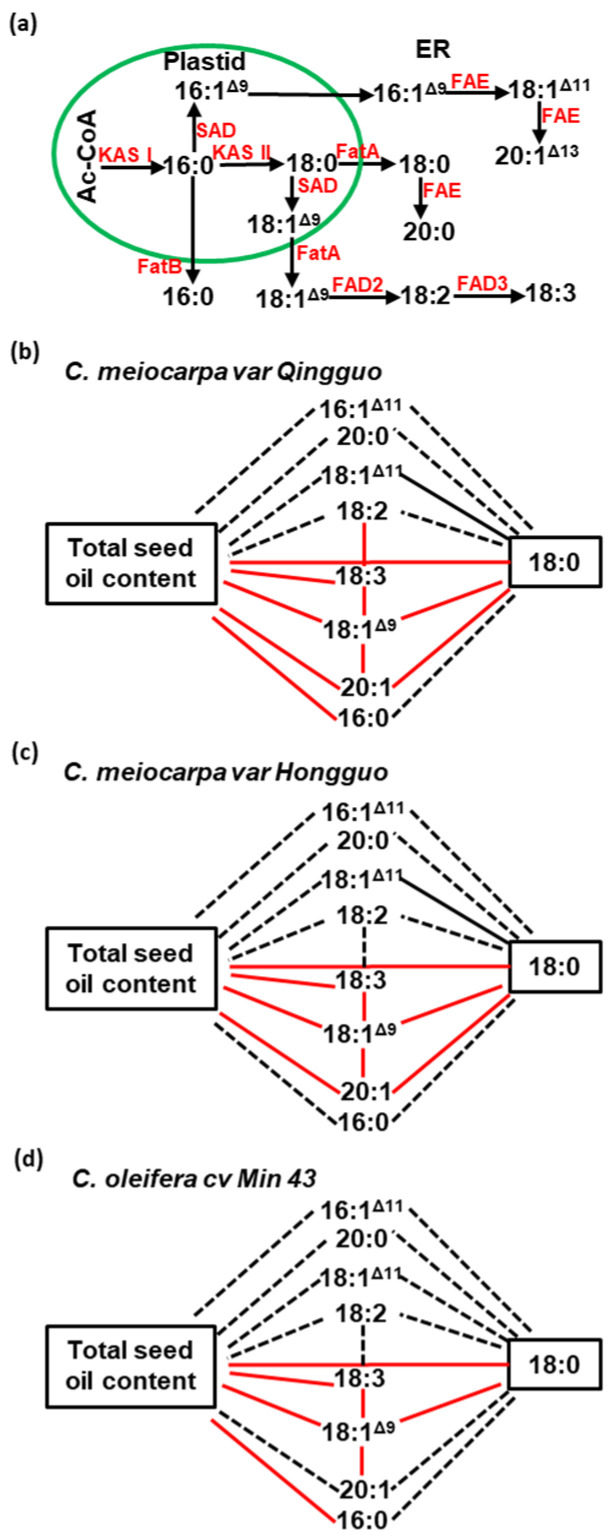
Correlations among seed oil contents and fatty acids. (**a**) Schematic representation of the fatty acid biosynthesis in plastid and endoplamic reticulum. (**b**) Correlations among total seed oil contents and individual fatty acids in *C. meiocarpa* var. *Qingguo*. (**c**) Correlations among total seed oil contents and individual fatty acids in *C. meiocarpa* var. *Hongguo*. (**d**) Correlations among total seed oil contents and individual fatty acids in *C. oleifera* cv. *Min 43*. Solid red lines represent significant positive correlations; solid black lines represent significant negative correlations; black dash lines represent insignificant correlations.

**Figure 6 plants-12-02591-f006:**
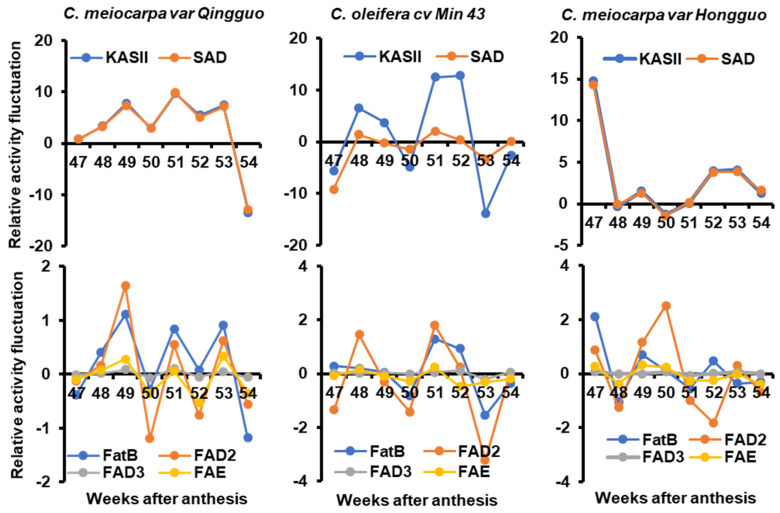
The relative enzyme activity fluctuations in *C. meiocarpa* var. *Qingguo* (**left panel**), *C. oleifera* cv. *Min 43* (**middle panel**), and *C. meiocarpa* var. *Hongguo* (**right panel**) during seed desiccation.

**Figure 7 plants-12-02591-f007:**
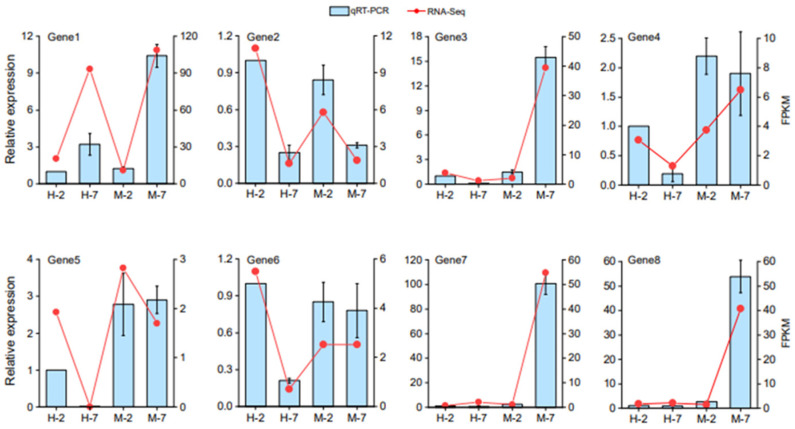
The transcriptional levels of eight candidate genes at early and late seed desiccation stages in *C. oleifera* cv. *Min 43* and *C. meiocarpa* var. *Hongguo*. The qRT-PCR data are shown in the left y-axis, RNA-Seq data (FPKM) are shown in the right y-axis. The H-2 and H-7 represent *C. meiocarpa* var. *Hongguo* seeds harvested on the 47th and 52nd week after anthesis, respectively; M-2 and M-7 represent *C. oleifera* cv. *Min 43* seeds harvested on the 47th and 52nd week after anthesis, respectively. *CoGAPDH* was used as reference gene for qRT-PCR, the relative expression level of H-2 sample was set as 1.

**Table 1 plants-12-02591-t001:** Differentially expressed genes between *Camellia oleifera* cv. Min 43 and *Camellia meiocarpa* var. *Hongguo* at the late stage of seed desiccation.

Gene ID in Hongguo	Gene ID in Min 43	KO Name	KEGG Description	Expression Fold Changes within Cultivar (1 November vs. 26 September)		Expression Fold Changes on 1 November
				Hongguo	Min 43	Min 43 Relative to Hongguo
Group I: Upregulated in both HG and Min 43						
TRINITY_DN16539_c0_g1_i1_2	TRINITY_DN24218_c0_g1_i2_1	ko00592	alpha-Linolenic acid metabolism	3.03	8.20	2.71
TRINITY_DN28162_c1_g1_i1_1	TRINITY_DN30108_c1_g2_i3_1	ko00592	alpha-Linolenic acid metabolism	2.62	32.56	12.43
TRINITY_DN28042_c0_g1_i1_2	TRINITY_DN30108_c0_g1_i1_1	ko00592	alpha-Linolenic acid metabolism	2.30	6.97	3.03
TRINITY_DN20394_c0_g1_i1_2	TRINITY_DN24587_c0_g1_i1_1	ko00010/ko00040/ko00561	Glycolysis/gluconeogenesis/pentose and glucuronate interconversions/glycerolipid metabolism	4.58	9.97	2.17
TRINITY_DN10717_c0_g1_i1_2	TRINITY_DN25787_c1_g1_i1_1	ko00062/ko04626	Fatty acid elongation/plant–pathogen interaction	8.62	~	~
TRINITY_DN21443_c0_g1_i1_2	TRINITY_DN32311_c1_g3_i3_2	ko01040	Biosynthesis of unsaturated fatty acids	2.01	5.79	2.88
Group II: Downregulated in both HG and Min 43						
TRINITY_DN25522_c0_g1_i1_1	TRINITY_DN32370_c0_g1_i4_2	ko00592	alpha-Linolenic acid metabolism	0.15	0.33	2.21
TRINITY_DN16629_c0_g1_i1_1	TRINITY_DN23517_c0_g1_i1_2	ko00010/ko00071/ko00350	Glycolysis/gluconeogenesis/fatty acid degradation/tyrosine metabolism	0.06	0.20	3.31
TRINITY_DN27681_c1_g3_i1_1	TRINITY_DN35019_c2_g1_i6_2	ko00561/ko00564	Glycerolipid metabolism/glycerophospholipid metabolism	0.21	0.47	2.21
TRINITY_DN846_c0_g1_i1_1	TRINITY_DN33688_c2_g1_i1_2	ko00561	Glycerolipid metabolism	0.00	0.11	~
TRINITY_DN29928_c0_g2_i4_1	TRINITY_DN27234_c0_g1_i1_1	ko00061/ko00254|ko00620/ko00640	Fatty acid biosynthesis/aflatoxin biosynthesis/pyruvate metabolism/propanoate metabolism	0.05	0.35	6.50
Group III: Downregulated in HG but upregulated in Min 43						
TRINITY_DN30040_c0_g3_i1_1	TRINITY_DN19234_c0_g1_i1_1	ko00073	Cutin, suberine, and wax biosynthesis	0.14	15.90	115.35
TRINITY_DN21501_c0_g1_i1_1	TRINITY_DN26190_c1_g1_i1_1	ko00100/ko00909	Steroid biosynthesis/sesquiterpenoid and triterpenoid biosynthesis	0.31	19.86	63.23
TRINITY_DN24679_c0_g2_i1_1	TRINITY_DN27816_c0_g1_i1_1	ko00564	Glycerophospholipid metabolism	0.39	2.47	6.32
TRINITY_DN26036_c4_g4_i3_1	TRINITY_DN29951_c2_g2_i5_1	ko00564	Glycerophospholipid metabolism	0.02	14.05	732.15
TRINITY_DN29303_c2_g1_i4_1	TRINITY_DN24438_c0_g1_i1_1	ko00561	Glycerolipid metabolism	0.47	2.24	4.81
TRINITY_DN29601_c0_g3_i9_1	TRINITY_DN30355_c1_g1_i2_1	ko00561	Glycerolipid metabolism	0.39	2.44	6.21
Group IV: Downregulated in HG but keep constant in Min 43						
TRINITY_DN19520_c0_g1_i1_1	TRINITY_DN2720_c0_g1_i1_1	ko00591	Linoleic acid metabolism	0.37	2.45	6.62
TRINITY_DN16030_c0_g1_i1_2	TRINITY_DN30055_c1_g2_i2_1	ko00592	alpha-Linolenic acid metabolism	0.33	1.03	3.07
TRINITY_DN29437_c0_g1_i10_1	TRINITY_DN35593_c1_g3_i1_2	ko00592	alpha-Linolenic acid metabolism	0.24	0.65	2.74
TRINITY_DN23188_c1_g1_i1_1	TRINITY_DN24343_c0_g2_i3_1	ko00071	Fatty acid degradation	0.00	0.60	~
TRINITY_DN29729_c0_g2_i1_1	TRINITY_DN23466_c2_g1_i2_2	ko00061/ko00071/ko04146	Fatty acid biosynthesis/fatty acid degradation/peroxisome	0.42	1.74	4.11
TRINITY_DN28894_c0_g2_i1_1	TRINITY_DN35399_c0_g3_i1_2	ko00061/ko00071/ko04146	Fatty acid biosynthesis/fatty acid degradation/peroxisome	0.25	0.52	2.08
TRINITY_DN25437_c0_g1_i1_1	TRINITY_DN30757_c0_g1_i1_1	ko00600	Sphingolipid metabolism	0.24	2.57	10.67
TRINITY_DN26102_c2_g4_i1_1	TRINITY_DN31693_c4_g3_i1_2	ko00511/ko00600	Other glycan degradation/sphingolipid metabolism	0.24	0.74	3.09
TRINITY_DN29059_c1_g1_i4_1	TRINITY_DN27160_c0_g1_i1_1	ko00564	Glycerophospholipid metabolism	0.46	1.86	4.03
TRINITY_DN23273_c1_g1_i1_1	TRINITY_DN28897_c0_g1_i4_2	ko00100	Steroid biosynthesis	0.13	1.00	7.75
TRINITY_DN27207_c0_g1_i3_1	TRINITY_DN27509_c2_g1_i11_1	ko00100	Steroid biosynthesis	0.36	0.76	2.11
TRINITY_DN20079_c0_g1_i2_1	TRINITY_DN20611_c0_g1_i1_1	ko00561	Glycerolipid metabolism	0.32	0.75	2.36
TRINITY_DN30241_c2_g1_i9_1	TRINITY_DN27804_c1_g2_i1_1	ko00564/ko00565/ko04144	Glycerophospholipid metabolism/ether lipid metabolism/endocytosis	0.33	2.10	6.39
TRINITY_DN17458_c1_g1_i1_1	TRINITY_DN30850_c1_g3_i1_2	ko00564/ko00565/ko04144	Glycerophospholipid metabolism/ether lipid metabolism/endocytosis	0.00	0.51	~
TRINITY_DN26926_c1_g2_i10_1	TRINITY_DN32652_c2_g2_i6_2	ko00564/ko00565	Glycerophospholipid metabolism/ether lipid metabolism	0.25	0.57	2.23
Group V: Kept constant in HG but upregulated in Min 43						
TRINITY_DN11159_c0_	TRINITY_DN22144_c0_	ko00600	Sphingolipid metabolism	1.53	3.48	2.27
g1_i1_2	g1_i1_1					
TRINITY_DN15140_c0_g1_i1_1	TRINITY_DN23601_c0_g1_i2_1	ko00600	Sphingolipid metabolism	0.58	7.34	12.59
TRINITY_DN15365_c0_g1_i1_2	TRINITY_DN32115_c0_g1_i1_1	ko00062/ko04626	Fatty acid elongation/plant–pathogen interaction	2.44	~	~
TRINITY_DN18887_c0_g1_i1_1	TRINITY_DN27285_c0_g1_i2_1	ko00062/ko04626	Fatty acid elongation/plant–pathogen interaction	0.75	2.92	3.90
TRINITY_DN19430_c0_g1_i1_1	TRINITY_DN27285_c0_g2_i2_1	ko00062/ko04626	Fatty acid elongation/plant–pathogen interaction	0.98	4.30	4.40
TRINITY_DN25961_c1_g2_i1_2	TRINITY_DN27285_c0_g3_i1_1	ko00062/ko04626	Fatty acid elongation/plant–pathogen interaction	1.48	115.07	77.76
TRINITY_DN21049_c0_g1_i1_2	TRINITY_DN24633_c0_g1_i1_1	ko00062/ko01040	Fatty acid elongation/biosynthesis of unsaturated fatty acids	0.92	3.58	3.91
TRINITY_DN7374_c0_g1_i1_1	TRINITY_DN6506_c0_g1_i1_1	ko00062/ko01040	Fatty acid elongation/biosynthesis of unsaturated fatty acids	1.05	3.00	2.85
TRINITY_DN17930_c0_g1_i1_1	TRINITY_DN21306_c0_g1_i1_1	ko00564/ko00565/ko04144	Glycerophospholipid metabolism/ether lipid metabolism/endocytosis	0.83	6.11	7.33
TRINITY_DN20368_c0_g1_i1_2	TRINITY_DN24269_c0_g1_i2_1	ko00564	Glycerophospholipid metabolism	1.58	3.29	2.08
TRINITY_DN22290_c0_g2_i1_2	TRINITY_DN26386_c2_g5_i1_1	ko00564	Glycerophospholipid metabolism	0.56	2.16	3.85
TRINITY_DN28036_c0_g2_i3_1	TRINITY_DN28544_c0_g1_i5_1	ko00564	Glycerophospholipid metabolism	0.76	2.10	2.77
TRINITY_DN18298_c0_g1_i1_1	TRINITY_DN22374_c0_g1_i1_1	ko00561/ko00564	Glycerolipid metabolism/glycerophospholipid metabolism	1.18	2.85	2.41
TRINITY_DN20304_c0_	TRINITY_DN20388_c0_	ko00561	Glycerolipid metabolism	1.78	3.65	2.05
g1_i1_1	g1_i2_1					
TRINITY_DN23410_c0_g1_i1_1	TRINITY_DN23078_c0_g1_i2_1	ko00561	Glycerolipid metabolism	0.67	2.51	3.73
TRINITY_DN27117_c0_g1_i6_2	TRINITY_DN30251_c2_g1_i2_1	ko00561/ko00564/ko04070	Glycerolipid metabolism/glycerophospholipid metabolism/phosphatidylinositol signaling system	1.91	4.17	2.19
TRINITY_DN27969_c6_g1_i5_2	TRINITY_DN29893_c0_g3_i1_1	ko00561	Glycerolipid metabolism	0.93	2.46	2.64
TRINITY_DN20974_c0_g1_i1_2	TRINITY_DN20019_c0_g1_i1_1	ko00592	alpha-Linolenic acid metabolism	1.26	4.43	3.51
TRINITY_DN24038_c0_g1_i1_2	TRINITY_DN27084_c0_g1_i2_1	ko00592	alpha-Linolenic acid metabolism	1.39	3.20	2.31
TRINITY_DN25575_c0_g2_i1_1	TRINITY_DN23850_c1_g1_i1_1	ko00480/ko00590	Glutathione metabolism/arachidonic acid metabolism	0.72	7.94	11.00
TRINITY_DN28455_c1_g1_i7_2	TRINITY_DN30495_c3_g2_i2_1	ko00073	Cutin, suberine, and wax biosynthesis	1.13	11.66	10.36
TRINITY_DN27207_c0_g1_i3_1	TRINITY_DN30341_c0_g1_i3_1	ko00100/ko00909	Steroid biosynthesis/sesquiterpenoid and triterpenoid biosynthesis	2.89	50.80	17.55
TRINITY_DN18365_c0_g2_i1_1	TRINITY_DN30341_c0_g2_i2_1	ko00100/ko00909	Steroid biosynthesis/sesquiterpenoid and triterpenoid biosynthesis	0.61	15.65	25.68
TRINITY_DN26549_c2_g1_i1_1	TRINITY_DN26052_c3_g2_i1_1	ko00100	Steroid biosynthesis	1.12	11.09	9.89
TRINITY_DN28994_c1_g1_i1_1	TRINITY_DN30362_c1_g2_i13_1	ko00100	Steroid biosynthesis	0.64	2.03	3.17
TRINITY_DN4285_c0_g1_i1_2	TRINITY_DN21633_c0_g1_i1_1	ko00100	Steroid biosynthesis	1.18	29.87	25.32

## Data Availability

The original contributions presented in the study are included in the article/Appendix A. Further inquiries can be directed to the corresponding author.

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
