# Peer review of "Integrative Metabolic and Transcriptomic Profiling in Camellia oleifera and Camellia meiocarpa Uncover Potential Mechanisms That Govern Triacylglycerol Degradation during Seed Desiccation"

_plants, 2023, doi:10.3390/plants12142591_

Round 1
Reviewer 1 Report
Authors have performed lipid and transcriptomic analysis of three Camellia sp. currently and newly cultivated in Chine for oil production. In order to identify the factors affecting Camellia oil synthesis, seeds were harvested weekly from late September to middle November, and seed oil contents and compositions were then measured and compared. Transcriptomics profiling at the early and the late desiccation stages, were correlated with genes associated to lipid metabolic pathway and provide novel insights about the potential mechanisms responsible for seed oil biosynthesis and degradation in this crop.
Authors have presented a nice work with robust and complete analysis on two species of Camellia, and provided validated new molecular markers of oil traits. To facilitate the comprehension of the manuscript, here some recommendations and comments.
Suggestions
- In the main text of results, it may be important to indicate that the plants were grown in the same location, and the difference in color and maturation it cannot be due to environmental conditions.
- Harvesting was done at different times every week for each specie. It is difficult to understand how authors have done to standardize the maturation stage of each fruit, supposing that fruits at different developmental stages are growing at the same time on the tree. Have the flowers been tagged and then considering fruits at a specific time after anthesis or the harvesting was done only base on the time?
- The two sections 2.2. The accumulation patterns of individual fatty acid species during seed desiccation and 2.3. The fatty acid profile changes during seed desiccation are redundant, and one of them should be removed.
- It could be interesting to perform a GO ontology on the all DEGs to evaluate the contribution of the genes involved in seed oil degradation compared to the rest of the DEGs.
- The seed oil accumulation from C. meiocarpa var Hongguo was much slower compared with C. meiocarpa var Qingguo or C. oleifera cv Min 43. Although transcriptomics analyses were performed only for the last two species, it could be interesting to provide qRT-PCR analysis on the three species for some candidates genes, to validate their implication in seed oil accumulation.
- Should conclusion go on sections 4, before M&M?
Errors.
Line 39 compounds including polyphenol, squalene, and vitamin E et al. (Liu et al., 2020), which
Line 134 Figure 2. Individual fatty acid content changes during seed disiccation in C. meiocarpa
Line 144 saw in HG (Figure 2C). The linoleic acid (18:2) is the major polyunsaturated fatty acid, Sometimes minuscule and sometimes capitals
Line 176 reached to nadir on Oct. 18, followed by increasing until Nov. 14. In contrast, QG and M43
Line 404 970). Besides facilitating seed germination, the soponins synthesized at late seed desic
Author Response
Authors have performed lipid and transcriptomic analysis of three Camellia sp. currently and newly cultivated in China for oil production. In order to identify the factors affecting Camellia oil synthesis, seeds were harvested weekly from late September to middle November, and seed oil contents and compositions were then measured and compared. Transcriptomics profiling at the early and the late desiccation stages, were correlated with genes associated to lipid metabolic pathway and provide novel insights about the potential mechanisms responsible for seed oil biosynthesis and degradation in this crop.
Authors have presented a nice work with robust and complete analysis on two species of Camellia, and provided validated new molecular markers of oil traits. To facilitate the comprehension of the manuscript, here some recommendations and comments.
Suggestions
- In the main text of results, it may be important to indicate that the plants were grown in the same location, and the difference in color and maturation it cannot be due to environmental conditions.
Authors’ response: thank you very much for your nice suggestion! We add this information at the beginning of the result section (line 102-104).
- Harvesting was done at different times every week for each specie. It is difficult to understand how authors have done to standardize the maturation stage of each fruit, supposing that fruits at different developmental stages are growing at the same time on the tree. Have the flowers been tagged and then considering fruits at a specific time after anthesis or the harvesting was done only base on the time?
Authors’ response: That’s a great question! We noticed that majority of the flowers started to open at similar time, so we decided not to tag flowers in the previous year. When harvesting fruit at each week, the fruits were selected based on the following criterions: 1) the fruits should be positioned on the same side of the tree, such that they likely receive similar amounts of sun light; 2) the fruits are similar in size and fruit color appearance for each species. We added this information into the material and method section (line 520-523).
- The two sections 2.2. The accumulation patterns of individual fatty acid species during seed desiccation and 2.3. The fatty acid profile changes during seed desiccation are redundant, and one of them should be removed.
Authors’ response: the section 2.2. is based on the absolute contents of each fatty acid (g FAME per 100 g dry seed mass); the section 2.3. is based on the Mol% which is relative quantification, which reflect oil compositional changes, these two sections describe seed fatty acid changes from different prospective. In response to your suggestion, we removed the original 2.3 section title (line 176), and merge them into one section, thank you!
- It could be interesting to perform a GO ontology on the all DEGs to evaluate the contribution of the genes involved in seed oil degradation compared to the rest of the DEGs.
Authors’ response: Nice suggestion,thanks! We add these contents in the main text (lines 307-316), and supplied these data as new Supplementary table S10-S11 and new Supplementary Figure S5-S6.
- The seed oil accumulation from C. meiocarpa var Hongguo was much slower compared with C. meiocarpa var Qingguo or C. oleifera cv Min 43. Although transcriptomics analyses were performed only for the last two species, it could be interesting to provide qRT-PCR analysis on the three species for some candidates genes, to validate their implication in seed oil accumulation.
Authors’ response: As we showed in Figure 1, before the 50th week the oil contents from C. meiocarpa var Qingguo were much lower compared with C. meiocarpa var Hongguo and C. oleifera cv Min 43, this is something unexpected considering that QG seeds reached maturity earlier as indicated by its earlier seed coat lignification and darkening (Supplementary Figure S2). On other hand, this might explain why QG seeds showed slower oil accumulation at the early phase of seed desiccation: the early seed coat lignification in QG could restrict photo assimilate transportation into seed kernel for oil synthesis. For this reason, in this study we exluded C. meiocarpa var Qingguo for further transcriptomics analyses.
- Should conclusion go on sections 4, before M&M?
Authors’ response: we move the conclusion section before M & M, thank you!
Errors.
Line 39 compounds including polyphenol, squalene, and vitamin E et al. (Liu et al., 2020), which
Authors’ response: revised, thanks!
Line 134 Figure 2. Individual fatty acid content changes during seed disiccation in C. meiocarpa
Authors’ response: Corrected (line 151), thanks!
Line 144 saw in HG (Figure 2C). The linoleic acid (18:2) is the major polyunsaturated fatty acid, Sometimes minuscule and sometimes capitals
Authors’ response: corrected (line 147).
Line 176 reached to nadir on Oct. 18, followed by increasing until Nov. 14. In contrast, QG and M43
Authors’ response: revised (line 198-200), thanks!
Line 404 970). Besides facilitating seed germination, the soponins synthesized at late seed desic
Reviewer 2 Report
This paper studies the factors affecting seed triacylglycerol synthesis in seeds of different varieties of the tea plant (Camellia). These varieties, named M43, QG and HG, have a different behaviour in their oil accumulation curve, especially in the final stages of development, when the varieties M43 and QG experience a decrease in fat content.
The authors approached the research by performing an extensive study of the oil accumulation curves followed by monitoring the expression of different genes related to lipid synthesis and derived metabolites. They tried to establish the metabolic differences between the three camellia lines based on such expression data.
The article is of interest, both basic and applied, and fits within the scope of the journal plants. In general, the use of English is adequate and the paper is well structured. In my opinion the introduction correctly explains the background of the paper, and is documented with adequate references.
As for the methods, results and discussion I have some concerns that make the manuscript not acceptable in its present form.
1.-Data of oil content. The figure depicting the oil content at different times (Figure 1) is confusing and did not represent the typical oil accumulation curve. I assume this is because the samples were taken at an advanced stage of seed development. Could the authors include a complete curve covering the entire period of seed formation? If not, could they cite a reference where this is described? Also, the X-axis notation is not appropriate for a plant biology article. Generally, the advancement of seed maturation is given as days or weeks after flowering. Would it be possible to change the notation?
2.-Figure 2 with the fatty acid content should be deleted and replaced by a composition figure with a larger scale for each fatty acid. The composition in percentage is a much more reliable and stable value than the content per seed. On the other hand, the scale in Figure 3 does not make it possible to see the values corresponding to the minority fatty acids. Table 2 has no units on the X-axis.
3.- The variation in the oleic/linoleic ratio has been described in other oil crops and may be due to changes in ambient temperature. We must consider that the plants were not grown under controlled temperature, so changes in ambient temperature could be responsible for the variation of these fatty acids. Do the authors have the maximum and minimum temperatures at the points where change was observed?.
4- There are no units marked on the X-axis in figures 3 and 4.
5.- The sampling and statistical study applied to the composition data is not well explained. It is mentioned that n=3, but no further information is given. 3 what? 3 seeds? 3 batches of seeds? The authors should explain how the sampling was carried out in more detail in the materials and methods section. This question is important to determine if the differences in composition are significant.
6.-Results. The non-appearance of transcripts for enzymes related to triacylglyceride degradation such as lipases among the candidate genes for determining the low oil phenotype is surprising. This issue should be included in the discussion.
7.- please check all abbreviations. Some are not defined and others are cited differently throughout the manuscript. Thus line M43 sometimes appears as Min 43. Please homogenise.
In general the quality of English Language is acceptable.
Author Response
Reviewer 2:
This paper studies the factors affecting seed triacylglycerol synthesis in seeds of different varieties of the tea plant (Camellia). These varieties, named M43, QG and HG, have a different behaviour in their oil accumulation curve, especially in the final stages of development, when the varieties M43 and QG experience a decrease in fat content.
The authors approached the research by performing an extensive study of the oil accumulation curves followed by monitoring the expression of different genes related to lipid synthesis and derived metabolites. They tried to establish the metabolic differences between the three camellia lines based on such expression data.
The article is of interest, both basic and applied, and fits within the scope of the journal plants. In general, the use of English is adequate and the paper is well structured. In my opinion the introduction correctly explains the background of the paper, and is documented with adequate references.
As for the methods, results and discussion I have some concerns that make the manuscript not acceptable in its present form.
1.-Data of oil content. The figure depicting the oil content at different times (Figure 1) is confusing and did not represent the typical oil accumulation curve. I assume this is because the samples were taken at an advanced stage of seed development. Could the authors include a complete curve covering the entire period of seed formation? If not, could they cite a reference where this is described? Also, the X-axis notation is not appropriate for a plant biology article. Generally, the advancement of seed maturation is given as days or weeks after flowering. Would it be possible to change the notation?
Authors’ response: Thank you very much for your thoughtful comments! The purpose of this research is trying to understand the factors affecting Camellia seed oil accumulation during the advanced stages of seed development, namely seed desiccation stages. Thus, we did not take samples at the early phase of seed development. The Camellia seed oil accumulation have been investigated before, here we added two references which show a complete curve covering the entire period of Camellia seed oil formation.
Based on flowering date and the harvest date, we also convert the X-axis notation as weeks after anthesis, then replot Figure 1 to Figure 6.
2.-Figure 2 with the fatty acid content should be deleted and replaced by a composition figure with a larger scale for each fatty acid. The composition in percentage is a much more reliable and stable value than the content per seed. On the other hand, the scale in Figure 3 does not make it possible to see the values corresponding to the minority fatty acids. Table 2 has no units on the X-axis.
Authors’ response: We replot Figure 2 as composition figure, and units on the X-axis were added; the Figure 3 also was replotted to make it larger; the fatty acid content data also was supplied as new Supplementary Table S1, thank you!
3.- The variation in the oleic/linoleic ratio has been described in other oil crops and may be due to changes in ambient temperature. We must consider that the plants were not grown under controlled temperature, so changes in ambient temperature could be responsible for the variation of these fatty acids. Do the authors have the maximum and minimum temperatures at the points where change was observed?
Authors’ response: I agree with the reviewer that the variation in ambient temperature could affect oleic/linoleic ratio as documented in other oil crops. Unfortunately, we did not record the temperature at each sampling week.
4- There are no units marked on the X-axis in figures 3 and 4.
Authors’ response: Added, thanks!
5.- The sampling and statistical study applied to the composition data is not well explained. It is mentioned that n=3, but no further information is given. 3 what? 3 seeds? 3 batches of seeds? The authors should explain how the sampling was carried out in more detail in the materials and methods section. This question is important to determine if the differences in composition are significant.
Authors’ response: We used 3 batches of seeds as 3 replicates, each batch contain 20 seeds that are randomly mixed from same germplasm. We add these details in the materials and methods section (line 525-526).
6.-Results. The non-appearance of transcripts for enzymes related to triacylglyceride degradation such as lipases among the candidate genes for determining the low oil phenotype is surprising. This issue should be included in the discussion.
Authors’ response: Great question! More discussions were added regarding these point (line 460-469).
7.- please check all abbreviations. Some are not defined and others are cited differently throughout the manuscript. Thus line M43 sometimes appears as Min 43. Please homogenise.
Authors’ response: Checked and corrected through the manuscript.
Comments on the Quality of English Language
In general the quality of English Language is acceptable.
Authors’ response: Thank you very much for your positive encouragement!
Reviewer 3 Report
In this paper, the authors have performed a combination of lipid analysis (oil content and fatty acid composition) together with gene expression studies to analyze seed oil composition in three different germlines of Camellia. This plant species has both agronomic and biotechnological interests because of the edible use of its oil. The specific objective of the authors was to identify the molecular factors responsible of seed oil degradation in the late stages of seed maturation.
Overall, the paper is interesting and most part of the results support the conclusions from the authors. However, I have some questions that might require the attention of the authors before the paper is publishable:
1. One question that should be modified or addressed properly is the “activities” shown in Figure 5. In my opinion, activities cannot be monitored in the way the authors propose. The data shown in Fig.5 illustrate the changes in fatty acid composition occurring among the different days of harvesting and dessication. It is therefore a relative measurement that under no way represents the actual activity of the enzymes. By doing this the authors are making some assumptions that might not be correct. First one; while monitoring 18:2 or 18:3 fatty acid levels and correlating that with FAD2 or FAD3 activities, they are assuming that plastidial desaturases do not contribute to omega-6 or omega-3 fatty acid content. This could be true but it is an assumption that should be reinforced by some determinations. This is more relevant if we take into account that the authors mention that exocarp photosynthesis may contribute to Camelia seed oil accumulation. In my opinion this figure 5 should be removed because is confusing and does not reflect any actual data.
2. The correlations performed by the authors are also confusing. In my opinion, showing the seed oil content and the fatty acid composition at the different harvesting dates is clear enough to illustrate how these two parameters were modified. I should not need any further correlation analysis.
3. In the same line, if an important point of the discussion is the contribution of FAD genes to the fatty acid composition of the Camellia oil, I would strongly recommend that most if not all of these genes (FAD2, FAD3, FAB2, Estearoyl-ACP desaturase, FAD7, FAD6, FAD8…) should be included in the results of the paper. Authors have the RNASeq data and should confirm this by qPCR analysis. In addition to this, the figure S5 should be incorporated to the paper, not as supplementary data.
4. I missed some information of the number of replicates used in the RNASeq analysis. The same is true in other results of the paper. This should be explicitly mentioned in the methods section.
5. Figure 2. I should recommend the authors to revise this figure to try to show the data in other way. Readers can be confused by the split of the different fatty acid composition data into three subfigures for each variety. I see that the molar percentage is different and that if we represent them altogether some of the changes would not be visible. Maybe a graph with those more relevant a Table with all the data should be an idea.
very minor changes required. P 15, L. 332: consistent. Just check again fo minor spelling errors
Author Response
Reviewer 3
In this paper, the authors have performed a combination of lipid analysis (oil content and fatty acid composition) together with gene expression studies to analyze seed oil composition in three different germlines of Camellia. This plant species has both agronomic and biotechnological interests because of the edible use of its oil. The specific objective of the authors was to identify the molecular factors responsible of seed oil degradation in the late stages of seed maturation.
Overall, the paper is interesting and most part of the results support the conclusions from the authors. However, I have some questions that might require the attention of the authors before the paper is publishable:
- One question that should be modified or addressed properly is the “activities” shown in Figure 5. In my opinion, activities cannot be monitored in the way the authors propose. The data shown in Fig.5 illustrate the changes in fatty acid composition occurring among the different days of harvesting and dessication. It is therefore a relative measurement that under no way represents the actual activity of the enzymes. By doing this the authors are making some assumptions that might not be correct. First one; while monitoring 18:2 or 18:3 fatty acid levels and correlating that with FAD2 or FAD3 activities, they are assuming that plastidial desaturases do not contribute to omega-6 or omega-3 fatty acid content. This could be true but it is an assumption that should be reinforced by some determinations. This is more relevant if we take into account that the authors mention that exocarp photosynthesis may contribute to Camelia seed oil accumulation. In my opinion this figure 5 should be removed because is confusing and does not reflect any actual data.
Authors’ response: You are right that the data presented in Fig. 6 is relative measurement in nature, we used their products (fatty acids) changes to indirectly reflect corresponding enzyme activity changes, thus they do not represent the actual activity of the enzymes. To make it more clear we changed the Y-axis as “relative activity fluctuation” to avoid potential confusion. In this figure, we are trying to see whether various enzymes from the fatty synthesis pathway are synchronized or not.
Since FAD2’s substrate 18:1Δ9 is produced by SAD which is a plastidial enzyme (Figure 5a), thus plastidial desaturases especially SAD family play critical role for omega-6 or omega-3 fatty acid content. We add discussion in lines 392-396. We also discussed other plastidial desaturases (FAD6, FAD7, FAD8) for seed oil accumulation in lines 399-407.
- The correlations performed by the authors are also confusing. In my opinion, showing the seed oil content and the fatty acid composition at the different harvesting dates is clear enough to illustrate how these two parameters were modified. I should not need any further correlation analysis.
Authors’ response: We have two goals for the correlation analysis: 1) trying to identify the most important fatty acid components that contribute to the seed oil yield within each Camellia germplasm; 2) the relationships between individual fatty acid component within each Camellia germplasm.
- In the same line, if an important point of the discussion is the contribution of FAD genes to the fatty acid composition of the Camellia oil, I would strongly recommend that most if not all of these genes (FAD2, FAD3, FAB2, stearoyl-ACP desaturase, FAD7, FAD6, FAD8…) should be included in the results of the paper. Authors have the RNASeq data and should confirm this by qPCR analysis. In addition to this, the figure S5 should be incorporated to the paper, not as supplementary data.
Authors’ response: Thank you very much for your thoughtful comments! In Arabidopsis several stearoyl-ACP desaturase (SAD) work redundantly in plastid to catalyze 18:0-ACP to 18:1-ACP, these SAD genes include FAB2, AAD1, AAD5 and AAD6. Thus, the SAD in Figure 5a and Figure 6 essentially represent multiple paralogs of acyl-ACP desaturase including FAB2, AAD1, AAD5 and AAD6. At the late phase of seed desiccation, oil (or fatty acid) synthesis likely is not the dominant factor to determine seed oil level, rather TAG degradation become the dominant event in the seed as evidenced by a ~quarter oil loss within 1-2 week time frame, that also could explain why these fatty acid synthesis genes were not detected as DEGs at this specific seed developmental stage.
Camellia is an 18:3 plant, its leaf galactolipid is rich in 18:3, FAD6, FAD7 and FAD8 play critical role for galactolipid desaturation in plastid. However, in Camellia seed, only trace amounts of 18:3 was detected (<0.5% of seed dry mass; Figure 2), while 16:2 and 16:3 were undetectable (Supplementary Figure S3). Thus, we reasoned that FAD6, FAD7 and FAD8 do not make significant contribution to Camellia seed oil synthesis. So far, the underlying mechanisms leading to FAD6/FAD7/FAD8 differential expression between leaves and seeds remains unclear. We add these points into the discussion section (lines 392-396, and lines 399-407).
The previous Supplementary Figure S5 was incorporated into main text as new Figure 7.
- I missed some information of the number of replicates used in the RNASeq analysis. The same is true in other results of the paper. This should be explicitly mentioned in the methods section.
Authors’ response: For fatty acid analysis we used three replicates; for RNA-Seq we used two biological replicates. We added this information into the method section (Line 546 and lines 563-565).
- Figure 2. I should recommend the authors to revise this figure to try to show the data in other way. Readers can be confused by the split of the different fatty acid composition data into three subfigures for each variety. I see that the molar percentage is different and that if we represent them altogether some of the changes would not be visible. Maybe a graph with those more relevant a Table with all the data should be an idea.
Authors’ response: We revised Figure 2 into composition figure, we also provided these data in Supplementary Figure S1, thank you!
Comments on the Quality of English Language
very minor changes required. P 15, L. 332: consistent. Just check again for minor spelling errors.
Authors’ response: Thank you very much for your positive comments! We checked and revised throughout the manuscript.
Round 2
Reviewer 2 Report
All concerns were adressed. Te manuscript is acceptable in its present form.
Reviewer 3 Report
I wish to thank the authors to their responses to my review. I think that may corrections have been succesfully answered.